# Emergence and Spread of the SARS-CoV-2 Variant of Concern Delta across Different Brazilian Regions

Ighor Arantes,[a,b] Felipe Gomes Naveca,[c,d] Tiago Gräf,[e] COVID-19 Fiocruz Genomic Surveillance Network, Fábio Miyajima,[f] Helisson Faoro,[g] Gabriel Luz Wallau,[h] Edson Delatorre,[i] Luciana Reis Appolinario,[a] Elisa Cavalcante Pereira,[a] Tainá Moreira Martins Venas,[a] Alice Sampaio Rocha,[a] Renata Serrano Lopes,[a] Marilda Mendonça Siqueira,[a] Gonzalo Bello,[b] Paola Cristina Resende[a]

[a]Laboratório de Vírus Respiratórios e Sarampo, Instituto Oswaldo Cruz (IOC), Fundação Oswaldo Cruz (FIOCRUZ), Rio de Janeiro, Brazil
[b]Laboratório de AIDS e Imunologia Molecular, Instituto Oswaldo Cruz (IOC), Fundação Oswaldo Cruz (FIOCRUZ), Rio de Janeiro, Brazil
[c]Laboratório de Ecologia de Doenças Transmissíveis da Amazônia (EDTA), Instituto Leônidas e Maria Deane, Fundação Oswaldo Cruz (FIOCRUZ), Manaus, Amazonas, Brazil
[d]Laboratório de Flavivírus, Instituto Oswaldo Cruz (IOC), Fundação Oswaldo Cruz (FIOCRUZ), Rio de Janeiro, Brazil
[e]Instituto Gonçalo Moniz, Fundação Oswaldo Cruz (FIOCRUZ), Salvador, Bahia, Brazil
[f]Analytical Competence Molecular Epidemiology Laboratory (ACME), Fundação Oswaldo Cruz (FIOCRUZ), Fortaleza, Ceará, Brazil
[g]Instituto Carlos Chagas (ICC), Fundação Oswaldo Cruz (FIOCRUZ), Curitiba, Paraná, Brazil
[h]Instituto Aggeu Magalhães, Fundação Oswaldo Cruz (FIOCRUZ), Recife, Pernambuco, Brazil
[i]Departamento de Biologia, Universidade Federal do Espirito Santo (UFES), Alegre, Espírito Santo, Brazil

**ABSTRACT** The SARS-CoV-2 variant of concern (VOC) Delta was first detected in India in October 2020. The first imported cases of the Delta variant in Brazil were identified in April 2021 in the southern region, followed by more cases in different regions during the following months. By early September 2021, Delta was already the dominant variant in the southeastern (87%), southern (73%), and northeastern (52%) Brazilian regions. This study aimed to understand the spatiotemporal dissemination dynamics of Delta in Brazil. To this end, we employed a combination of maximum likelihood (ML) and Bayesian methods to reconstruct the evolutionary relationship of 2,264 VOC Delta complete genomes (482 from this study) recovered across 21 of the 27 Brazilian federal units. Our phylogeographic analyses identified three major transmission clusters of Delta in Brazil. The clade BR-I ($n = 1,560$) arose in Rio de Janeiro in late April 2021 and was the major cluster behind the dissemination of the VOC Delta in the southeastern, northeastern, northern, and central-western regions. The AY.101 lineage ($n = 207$) that arose in the Paraná state in late April 2021 and aggregated the largest fraction of sampled genomes from the southern region. Lastly, the AY.46.3 lineage emerged in Brazil in the São Paulo state in early June 2021 and remained mostly restricted to this state. In the rapid turnover of viral variants characteristic of the SARS-CoV-2 pandemic, Brazilian regions seem to occupy different stages of an increasing prevalence of the VOC Delta in their epidemic profiles. This process demands continuous genomic and epidemiological surveillance toward identifying and mitigating new introductions, limiting their dissemination, and preventing the establishment of more significant outbreaks in a population already heavily affected by the COVID-19 pandemic.

**IMPORTANCE** Amid the SARS-CoV-2 continuously changing epidemic profile, this study details the space-time dynamics of the emergence of the Delta lineage across Brazilian territories, pointing out its multiple introductions in the country and its most prevalent sublineages. Some of these sublineages have their emergence, alongside their genomic composition and geographic distribution, detailed here for the first time. A special focus is given to the emergence process of Delta outside the country's south and southeast regions, the most populated and subjects of most published SARS-CoV-2 studies in Brazil. In summary, the study allows a better comprehension of the evolution

Address correspondence to Paola Cristina Resende, paola@ioc.fiocruz.br.

The authors declare no conflict of interest.

process of a SARS-CoV-2 lineage that would be associated with a significant recrudescence of the pandemic in Brazil.

**KEYWORDS** COVID-19, SARS-CoV-2, variant of concern, Delta, B.1.617.2, Brazil

The severe acute respiratory syndrome coronavirus 2 (SARS-CoV-2) variant of concern (VOC) Delta (B.1.617.2+AY.*/G/478K.V1/21A) was first described in India in October 2020 (1). By April 2021, it was designated by the World Health Organization as a variant of interest (VOI), and on 11 May 2021, it was formally recognized as a VOC, as Delta possesses molecular signatures known or predicted to have phenotypic implications and worldwide epidemiological relevance (2). By July 2021, VOC Delta was the most represented lineage in the SARS-CoV-2 global pandemic profile (3). VOC Delta exhibits multiple molecular signatures in its Spike protein (T19R, E156G, Δ157-158, L452R, T478K, P681R, and D950N) (4), with immunological (5), clinical, and epidemiological implications (6, 7).

Brazil registered its first COVID-19 case in late February 2020 (8). Since then, the country has been heavily affected by the pandemic, with more than 20 million cases and 600 thousand confirmed deaths by early October 2021 (8). From February until June 2021, the epidemic profile across the country has been largely dominated by the VOC Gamma (P.1+P.1.*) (9), which emerged in the northern Brazilian state of Amazonas in late November 2020 (10, 11). Since June 2021, however, the VOC Delta has been occupying increasingly more significant shares of the country's epidemic profile (Fig. 1). By early September 2021, most Brazilian sampled genomes were assigned to the B.1.617.2 or AY.* PANGO lineages in the country's southeastern (87%), southern (73%), and northeastern (52%) regions, while VOC Gamma still dominates in the northern (86%), and central-western (79%) regions (9).

The first imported case of the VOC Delta in Brazil was identified in the southern region by late April 2021 in an event epidemiologically linked to a Brazilian with a travel history through Asia in previous weeks (12). This first case was one of the many independent VOC Delta introductions to be detected in the country during the following months, some already producing, by April 2021, evidence of autochthonous transmission (13). In July 2021, the first large local cluster of the Delta variant was detected in the southeastern state of Rio de Janeiro (14). Previous analyses of Delta in Brazil primarily focus on the most populated states of Rio de Janeiro and São Paulo. In contrast, information about the dissemination dynamics of the Delta variant in other states is scarce. In this work, we aimed to understand the emergence and spread of the VOC Delta in Brazil by inferring the variant's phylogenetic profile across the country's five geographical regions and determining its major dissemination routes.

## RESULTS

The maximum likelihood (ML) phylogenetic analysis of 2,264 Delta sequences from Brazil and a selected subset of 591 non-Brazilian sequences revealed a large number (≥60) of discernible independent introductions in Brazil, mainly in the southeastern and southern regions (Fig. 2A). Most introductions resulted in small clusters (2 to <20) that were composed almost exclusively of Brazilian sequences (≥99%), including the Brazilian cluster (n = 5) associated with the Delta index case in the state of Paraná (PR-I) (Fig. 2A and Table 1). Three founder events, in contrast, resulted in major clusters in Brazilian territories, the cluster BR-I (n = 1,560), composed of lineages AY.99.1 and AY.99.2, and the lineages AY.101 (n = 208), and AY.46.3 (n = 171) (Fig. 2A). Among all genome samples submitted to GISAID, 58% of the ones assigned to the AY.101 lineage and 99% in lineage AY.46.3 were derived from Brazil. Sampled genomes from Brazil's southeastern region make up the majority (≥90%) of BR-I and AY.46.3. In comparison, those from the southern region represent the largest part (≥75%) of AY.101. BR-I and AY.101 were detected in different Brazilian states, while AY.46.3 was mainly restricted to the São Paulo state (Table 1). BR-I was the most represented cluster among genomes from

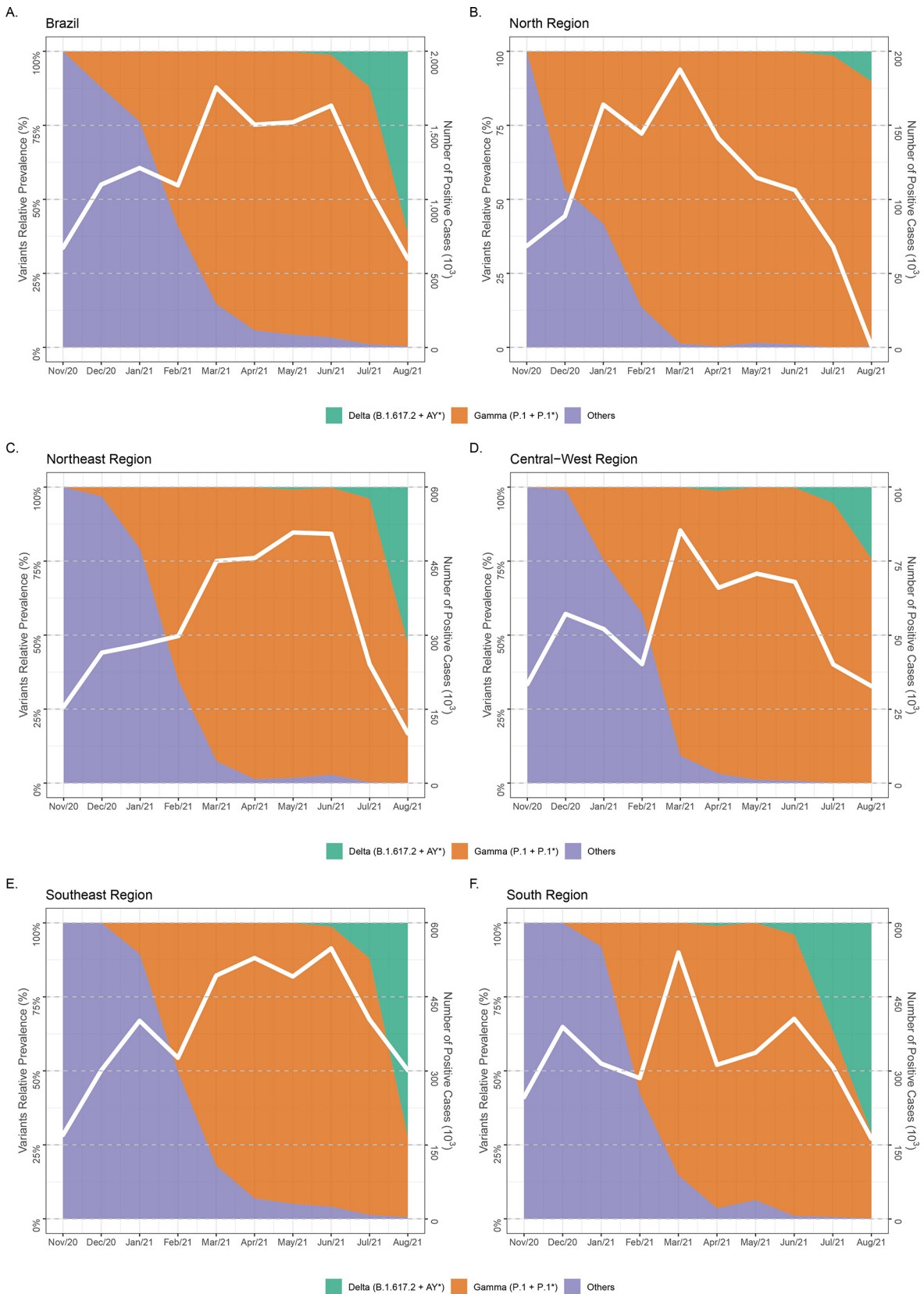

**FIG 1** Temporal evolution of the SARS-CoV-2 lineage composition in Brazil. (A to F) The graphs depict the relative prevalence of SARS-CoV-2 lineages among genomes sampled in Brazil and submitted to GISAID between late 2020 and mid-2021 represented both in the country level (A) and discriminated in its five geographic regions (B to F). The total of SARS-CoV-2-positive cases is also represented in each graph (A to F) for the geographic region in question.

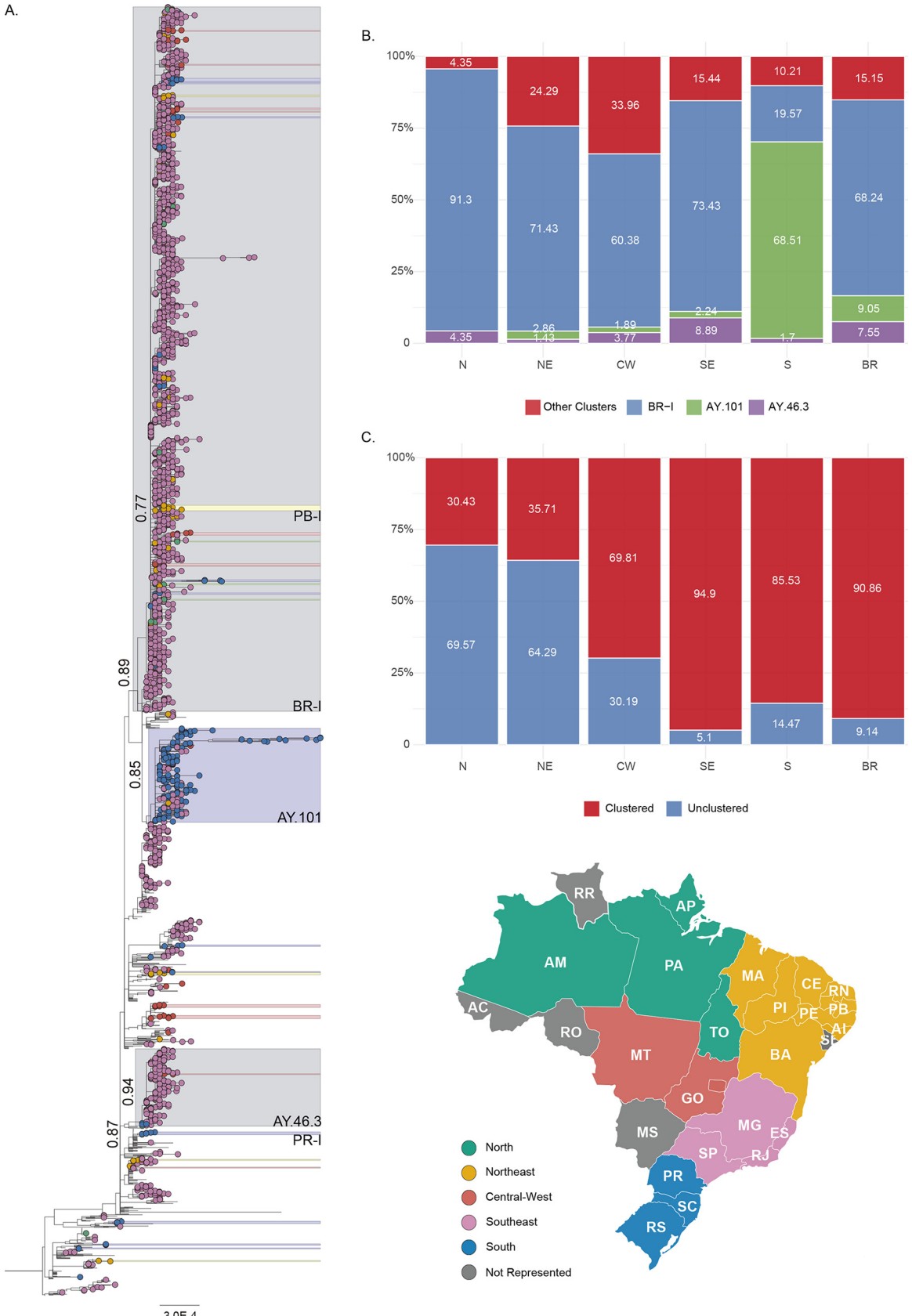

**FIG 2** Emergence of SARS-CoV-2 VOC Delta Brazilian phylogenetic clusters. (A) Maximum likelihood (ML) tree constructed with Brazilian SARS-Cov-2 VOC Delta (B.1.617.2 + AY*) genomes (*n* = 2,855). Tip shapes are colored according to the map in the bottom right of their

**TABLE 1** SARS-CoV-2 VOC Delta Brazilian data set[a]

| Location | | Amount No. | % | Sampling range (mo/yr) | Main amount No. | % | BR-I No. | % | AY.101 No. | % | AY.46.3 No. | % |
|---|---|---|---|---|---|---|---|---|---|---|---|---|
| N | AM | 18 | 0.80 | 07/21–08/18 | | | 18 | 1.17 | | | | |
| | AP | 3 | 0.13 | 08/03–08/05 | | | 3 | 0.19 | | | | |
| | PA | 1 | 0.04 | 07/28 | 1 | 0.29 | | | | | | |
| | TO | 1 | 0.04 | 07/13 | | | | | | | 1 | 0.58 |
| | TOTAL | 23 | 1.02 | 07/21–08/18 | 1 | 0.29 | 21 | 1.36 | | | 1 | 0.58 |
| NE | AL | 5 | 0.22 | 07/13–08/09 | | | 2 | 0.13 | 2 | 0.98 | 1 | 0.58 |
| | BA | 5 | 0.22 | 07/21–08/18 | | | 5 | 0.32 | | | | |
| | CE | 19 | 0.84 | 07/21–08/02 | 6 | 1.75 | 13 | 0.84 | | | | |
| | MA | 5 | 0.22 | 05/16 | 5 | 1.46 | | | | | | |
| | PB | 21 | 0.93 | 07/21–08/13 | | | 21 | 1.36 | | | | |
| | PI | 1 | 0.04 | 08/09 | | | 1 | 0.06 | | | | |
| | PE | 13 | 0.57 | 07/02–08/05 | 6 | 1.75 | 7 | 0.45 | | | | |
| | RN | 1 | 0.04 | 08/09 | | 0.00 | 1 | 0.06 | | | | |
| | TOTAL | 70 | 3.09 | 07/13–08/18 | 17 | 4.96 | 50 | 3.24 | 2 | 0.98 | 1 | 0.58 |
| CW | GO | 52 | 2.30 | 04/27–08/24 | 18 | 5.25 | 31 | 2.01 | 1 | 0.49 | 2 | 1.17 |
| | MT | 1 | 0.04 | 07/28 | | | 1 | 0.06 | | | | |
| | TOTAL | 53 | 2.34 | 04/27–08/24 | 18 | 5.25 | 32 | 2.07 | 1 | 0.49 | 2 | 1.17 |
| S | PR | 101 | 4.46 | 04/26–08/04 | 12 | 3.50 | | | 89 | 43.41 | | |
| | RS | 69 | 3.05 | 06/17–08/13 | 3 | 0.87 | 27 | 1.75 | 36 | 17.56 | 3 | 1.75 |
| | SC | 65 | 2.87 | 06/25–08/02 | 9 | 2.62 | 19 | 1.23 | 36 | 17.56 | 1 | 0.58 |
| | TOTAL | 235 | 10.38 | 04/26–08/13 | 24 | 7.00 | 46 | 2.98 | 161 | 78.54 | 4 | 2.34 |
| SE | ES | 20 | 0.88 | 07/05–08/06 | 2 | 0.58 | 18 | 1.17 | | | | |
| | MG | 32 | 1.41 | 05/22–08/12 | 3 | 0.87 | 29 | 1.88 | | | | |
| | RJ | 745 | 32.91 | 05/22–08/17 | 10 | 2.92 | 732 | 47.38 | 2 | 0.98 | 1 | 0.58 |
| | SP | 1086 | 47.97 | 05/22–08/23 | 268 | 78.13 | 617 | 39.94 | 39 | 19.02 | 162 | 94.74 |
| | TOTAL | 1883 | 83.17 | 05/22–08/13 | 283 | 82.51 | 1,396 | 90.36 | 41 | 20.00 | 163 | 95.32 |
| BR | TOTAL | 2264 | 100 | 05/22–08/23 | 343 | 100 | 1,545 | 100 | 205 | 100 | 171 | 100 |

[a]The table details the VOC Delta (B.1.617.2+AY*) data set (n = 2,264) of Brazilian sequences and their placement in a maximum likelihood topology. For each state, the sampling range and distribution of their sequences across the main tree and the three major Brazilian clusters (BR-I, AY.101, and AY.46.3) are detailed. CW, central-western; N, northern; NE, northeastern; S, southern; SE, southeastern; AM, Amazonas; AP, Amapá; PA, Pará; TO, Tocantins; AL, Alagoas; BA, Bahia; CE, Ceará; ES, Espírito Santo; MA, Maranhão; MG, Minas Gerais; PB, Paraíba; PI, Piauí; PE, Pernambuco; RJ, Rio de Janeiro; RN, Rio Grande do Norte; GO, Goiânia; MT, Mato Grosso; PR, Paraná; RS, Rio Grande do Sul; SC, Santa Catarina; SP, São Paulo.

Brazil's northern (91%), southeastern (73%), northeastern (71%), and central-western (60%) regions, and AY.101 was the most prevalent cluster in the southern region (69%) (Fig. 2B and Table S1 in the supplemental material).

Most Delta sequences sampled in the southeastern (95%) and southern (86%) regions branched within Brazilian clusters, while much lower levels of clustering were detected in the other regions (Fig. 2C and Table S2). The northern Brazilian region was the least represented in the national Delta data set (n = 23, 1%), harbored samples from four of the region's six states, and had the lowest level of local clustering (30%). Amazonas was the most represented state (n = 18, 0.8%) (Table 1), with samples collected since late July 2021, and displayed the largest Delta cluster in the region (n = 3 sequences). The northeastern region (n = 70.3%) harbored samples from eight of the region's nine states, the most represented ones being Paraíba (n = 21), Ceará (n = 19), and Pernambuco (n = 13) (Table 1). Despite a larger number of sequences than in Brazil's northern region, the clustering level observed (36%) was similar in both regions. The largest Delta cluster in the northeastern region was detected in the Paraíba state

**FIG 2** Legend (Continued)
Brazilian geographic region of origin. States not represented in the data set are colored in dark gray. All statistically supported (approximate likelihood-ratio test [aLRT] ≥ 0.75) groups independent of their dimensions have the same color scheme. Two of Brazil's main clusters (BR-I and AY.101), composed of southeastern sequences, are colored in light gray. Sequences outside Brazil had their tips removed for improved clarity. The three main clusters (BR-I, AY.101, and AY.46.3) and other noticeable clusters are named, and their statistical support (aLRT) is also indicated. (B) Distribution of sampled genomes from Brazil (BR, n = 2,264) and its North (N, n = 23), Northeast (NE, n = 70), Central-West (CW, n = 53), Southeast (SE, n = 1,883), and South (S, n = 235) regions across the three main clusters (BR-I, AY.101, and AY.46.3) and the main inferred ML VOC Delta tree (main). (C) Distribution of sampled genomes from Brazil and its six regions across the ML tree clustering profile, unclustered and clustered, a category defined by any statistically supported (aLRT ≥ 0.75) grouping of more than one sequence from the same geographical region. Maps were obtained from https://d-maps.com/.

(PB-I, *n* = 12 sequences) and comprises 57% of Delta sequences from that state (Fig. 2A). The central-west region (*n* = 53, 2%) is represented in the dataset by sequences from two of three states, being nearly all recovered from the Goiás state since late July (*n* = 52) (Table 1). The observed level of clustering in Goiás state (70%) was significantly higher than that in the northern and northeastern regions. However, Delta sequences were highly scattered in the tree's topology, forming multiple (*n* = 11) small clusters (2 to ≤7 sequences).

While the dissemination dynamics of Delta cluster BR-I was already explored in previous studies (13, 14), the spatiotemporal pattern of AY.101 and AY.46.3 in Brazil remained unknown. Bayesian phylogeographic analysis supports that Paraná was the most probable source of AY.101 (posterior state probability [PSP], 0.57) in late April 2021 (median, 27 April 2021; 95% highest probability density [HPD], 21 March 2021 to 17 May 2021) (Fig. 3A). Paraná was the main dissemination hub of cluster AY.101, followed by Santa Catarina. Most AY.101 genomes hosted the molecular synapomorphies ORF3a:A23V (100%), ORF1a: A3070V (98%), and S:T95I (85%) (Fig. 3C). The most recent common ancestor (MRCA) of cluster AY.46.3 in Brazil was traced back to early June 2021 (median, 4 June 2021; 95% HPD, 13 May 2021 to 20 June 2021) in the São Paulo state (PSP, 0.58) (Fig. 3B). São Paulo was the main hub of dissemination of the AY.46.3 lineage. Despite being detected in seven states, the dissemination of AY.46.3 outside São Paulo was highly limited and mostly reduced to single introductions, except for one small cluster in Goiás (*n* = 2; PSP, 1.0). AY.46.3 genomes had one molecular synapomorphy, ORF9b:R32L (Fig. 3C).

## DISCUSSION

In this work, we explored the emergence and spread of the VOC Delta in different Brazilian regions. The cluster BR-I, previously associated with the large local cluster of the Delta variant in the southeastern state of Rio de Janeiro (14), was here confirmed as the main driver behind the variant expansion in several Brazilian states of the northern, northeastern, and central-western regions. The Delta epidemic in the southern region, however, was mainly driven by the AY.101 lineage. A third transmission cluster, the AY.46.3 lineage, was also identified mainly in São Paulo.

The southeastern Brazilian region contributed 83% (*n* = 1,833) of the VOC Delta sequences analyzed here, followed by the southern region with 10% (*n* = 235), northeastern region with 3% (*n* = 70), central-western region with 2% (*n* = 53), and northern region with 1% (*n* = 23). This distribution mirrors the relative prevalence of VOC Delta across regions and closely resembles the inferred clustering levels. The proportion of sequences in each region with statistically supported association to another sampled genome from the same region was used as a proxy of local transmission events in opposition to independent introductions. The southeastern and southern regions displayed more sustained VOC Delta epidemics, with large fractions of their sampled genomes (86 to 95%) clustered among each region's data set. On the other hand, lower levels of clustering (30 to 70%) were observed in the remaining regions.

Our data support that Brazilian states present Delta epidemics in different stages of maturity. The oldest and most widely established Delta epidemics were observed in the southeastern and southern regions. The Delta epidemic in Goiás (central-west region), as in most of the states from the northeastern region, is currently driven by multiple small clusters, characteristic of an intermediate stage. The most notable exception was the state of Paraiba, which had the largest phylogenetic cluster of Delta in the northeastern region, and this cluster comprises more than 50% of the Delta sequences from that state. The Delta epidemic in the northern region seems to be at the very early dissemination stage, as most sequences from this region (70%) appeared unclustered, with almost no evidence of local spread. Continuous genomic monitoring in these states is essential, as these are likely candidates for becoming secondary hubs of VOC Delta dissemination in their respective regions.

The AY.101 lineage was the primary lineage responsible for disseminating the VOC Delta in Brazil's southern region, aggregating a significant fraction of its samples (69%), and was responsible for an even more significant fraction of samples from Paraná state

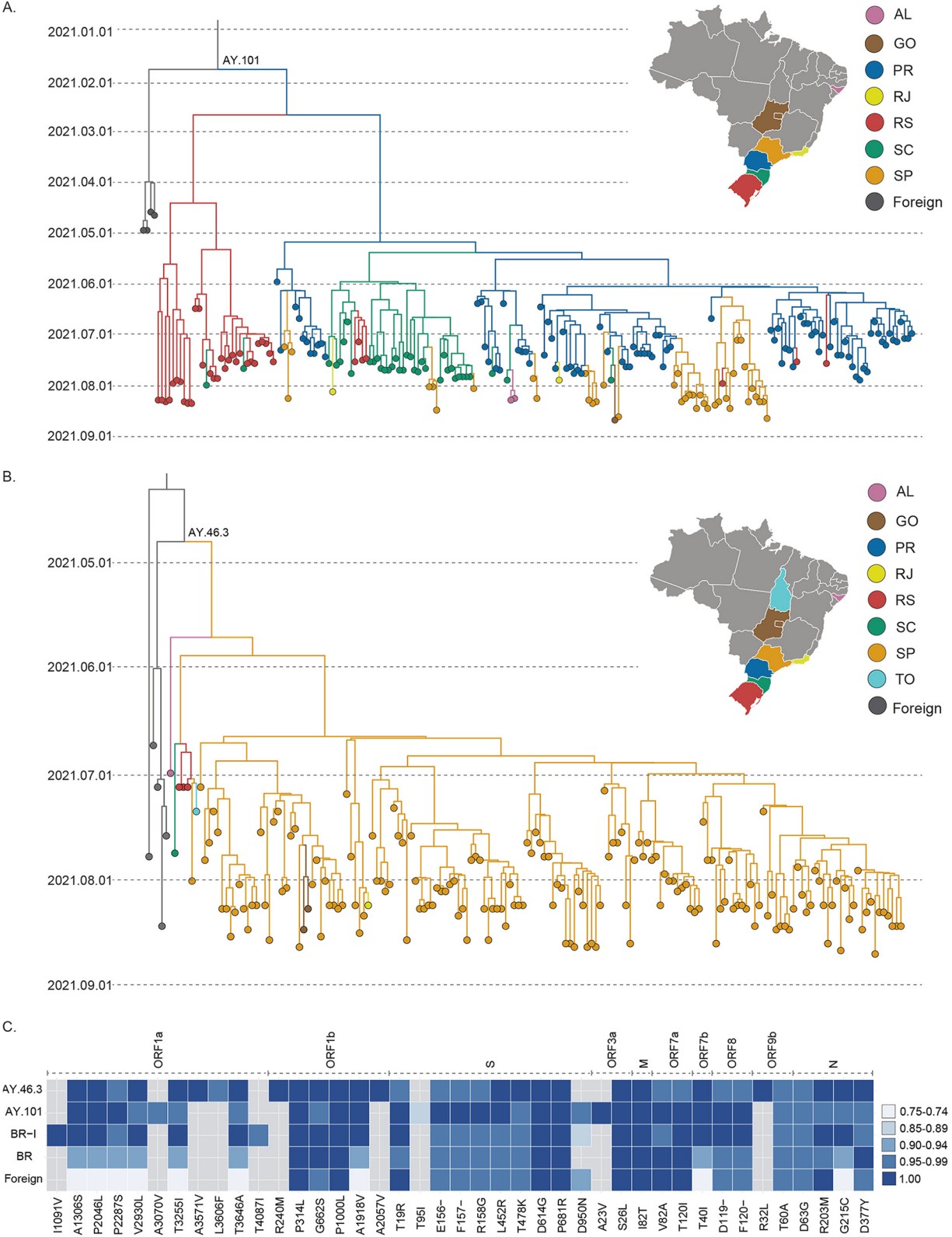

**FIG 3** Spatial, temporal, and molecular characterization of VOC Delta AY.101 and AY.46.3 lineages. (A and B) Time-scaled MCC tree of SARS-Cov-2 VOC Delta (B.1.617.2+AY*) AY.101 cluster (*n* = 207) (A) and AY.46.3 (*n* = 171) (B), one of the main Brazilian phylogenetic clusters of this variant. Branches are colored according to their most probable location, based on the color scheme shown in the upper-right corner. (C) Description of molecular signatures of the AY.101 lineage. Lines indicated by BR-I (*n* = 1,560), AY.101 (*n* = 208), and AY.46.3 (*n* = 171) summarize, in comparison to the Wuhan 2019 reference sequence, the relative frequency of nonsynonymous substitutions and deletions observed in the majority (≥75%)

(88%), the most probable source location. Other sampled genomes from Paraná were mostly aggregated in small clusters (*n* = 3), and no introduction of the cluster BR-I was detected in the state, even though BR-I has multiple sampled genomes in the other two southern states, Rio Grande do Sul (*n* = 27) and Santa Catarina (*n* = 19). This result could indicate that AY.101 dominates the VOC Delta epidemic in the Paraná state, limiting the spread of other clusters in the same network of susceptible individuals. Given that the MRCA of AY.101 was traced back to late April 2021, close to the first detection of the VOC Delta in the country, a founder event effect could be associated with the successful dissemination of this cluster across Brazil's southern region.

The AY.101 lineage exhibits ORF3a:A23V as a molecular synapomorphy. The ORF3a gene is an ion transporter of 275 amino acids (15), which when upregulated, leads to increased fibrinogen secretion associated with the characteristic COVID-19 cytokine storm (16). ORF3a activates the NLRP3 inflammasome by promoting TNF receptor-associated factor 3 (TRAF3)-mediated ubiquitination of apoptosis-associated speck-like protein with a caspase recruitment domain (17). The position identified here is located near the ORF3A:TRAF3 interaction site (18), raising the possibility of eventual phenotypic implications. Among the three sites (ORF1a:A3070V, S:T95I, and ORF3a:A23V) with higher prevalence in the AY.101 lineage, the presence of ORF3a:A23V is significantly distinct from other major Brazilian clusters and foreign reference sequences. The ORF3a:A23V mutation was a unique signature observed in all sequences within this cluster but was not found in the other genomes in our data set.

The AY.46.3 lineage was the most recently emerged of the three main Brazilian Delta clusters, having its MRCA traced back to early June. Most genomes sampled outside São Paulo (*n* = 6) were aggregated in the base of the cluster, which could be associated with the inferred low support for the AY.46.3 MRCA location in São Paulo (PSP, 0.58). Despite being heavily composed of sequences from São Paulo (≥90%), all Brazilian regions were represented in the AY.46.3 data set. The AY.46.3 lineage also exhibited one molecular synapomorphy, ORF9b:R32L. The ORF9b product localizes in the membrane of the mitochondria, acting in the inhibition of interferon-1 (IFN-1) secretion (19).

The clusters BR-I, AY.101, and AY46.3 make up the majority (85%) of Delta sequences analyzed here, while most other Brazilian sequences represented dead-end introductions or unsustained transmission clusters. Among these limited outbreaks was Brazil's initially identified cluster PR-I (*n* = 5), associated with the country's index case located in Paraná (13), and two additional limited outbreaks registered in Brazil's northeastern region. Both were associated with marine vessels: one in the state of Maranhão (*n* = 5 sequences), returning from Malaysia in mid-May 2021, and the other in the state of Pernambuco (*n* = 5 sequences) in early June 2021 (13). The lack of evidence of further transmission of these Delta clusters supports the success of preventive measures implemented in Brazil to contain such initial introductions.

It is interesting to note that dissemination of Brazilian Delta clades occurred after a large Gamma wave and during vaccination roll-out in Brazil. The time to the most recent ancestor ($T_{MRCA}$) of Brazilian Delta clades were traced back to between April and June 2021. Vaccination started in Brazil in February 2021, and around 65% and 25% of the country's population had already received one or two vaccine doses, respectively, by early September 2021 (Fig. S1). Thus, the high prevalence of population immunity (natural and/or vaccine-induced) in Brazil was not able to prevent the successful community dissemination of the VOC Delta and the consequent gradual replacement of the Gamma lineage across the country, consistent with the notion that VOC Delta is more transmissible than Gamma. In contrast to the VOC Gamma, however, the dissemi-

**FIG 3** Legend (Continued)

of sequences composing each cluster. The "BR" line represents the same procedure applied to sequences from Brazil outside the three main clusters (*n* = 325), and the "foreign" line to the ones outside Brazil is used as a reference in the complete ML tree (*n* = 591). The substitution and its gene are annotated in the bottom and top margins, respectively. Frequencies are represented according to the legend on the right. AL, Alagoas; GO, Goiás; PR, Paraná; RS, Rio Grande do Sul; SC, Santa Catarina; SP, São Paulo; TO, Tocantins. Maps were obtained from https://d-maps.com/.

nation of the different Delta sublineages was not associated with an exponential upsurge of SARS-CoV-2 cases in Brazil (Fig. 1).

This work is certainly limited by the nonavailability of more robust metadata and possibly insufficient sampling in several Brazilian states. VOC Delta is relatively susceptible to the available vaccines (20, 21). In our data set, five states were not represented by any sequence, Acre, Roraima, and Rondônia in the northern, Mato Grosso do Sul in the central-west, and Sergipe in the northeast. In contrast, other states were represented by very few ($n < 10$) sequences. A more globally strenuous sampling could lead to the elucidation of dissemination paths and identification of new clusters across Brazilian states with lower Delta prevalence. This effort would allow for more promptness in the response of public health authorities in their control of new introductions.

In the rapid turnover of variants characteristic of the SAR-CoV-2 pandemic, Brazilian regions seem to occupy different stages of an increasingly more significant participation of the VOC Delta in their epidemic profiles. This process demands continuous genomic and epidemiologic surveillance to identify and mitigate new introductions, limit their dissemination, and prevent the establishment of massive outbreaks in a population already heavily affected by the COVID-19 pandemic.

## MATERIALS AND METHODS

**Data set composition.** A multicentric effort from the COVID-19 FIOCRUZ Genomic Surveillance network in Brazil (9) recovered VOC Delta complete genomes ($n = 482$) across 20 of 26 Brazilian states. The majority of the SARS-CoV-2 VOC Delta (B.1.617.2+AY.*) genomes were newly generated using the Illumina COVIDSeq test kit with the adaptation of some primers to minimize regions with dropout (10) or previously published in-house sequencing protocols (22). Obtained FASTQ reads were imported into the CLC Genomics Workbench v.20.0.4 (Qiagen A/S, Denmark), trimmed, and mapped against a reference sequence (EPI_ISL_402124) available in the EpiCoV database of GISAID (23), or the consensus sequences were generated using ViralFlow (24). All genomes were uploaded to the EpiCoV database in GISAID (23) (Supplemental Material).

Additionally, we downloaded all VOC Delta (B.1.617.2+AY.*) complete genomes available in the EpiCoV database in GISAID (23) as of 7 September 2021 ($n = 948,925$). Sequences (i) without a complete collection, as well as (ii) those above a threshold of unidentified positions (Ns > 3% for Brazilian ones and Ns > 1% for those outside the country) were removed. The VOC Delta Brazilian sequences finally selected ($n = 2,264$) were combined with a subset of non-Brazilian Delta sequences selected after two sequential steps: we first removed identical sequences by their clustering with CD-HIT v.4.8.1 (25), and subsequently, we selected the 10 most similar hits to each queried Brazilian sequence by a local BLAST search (26). The resulting complete data set of Brazilian ($n = 2,264$) and non-Brazilian sequences ($n = 591$) was aligned with MAFFT v.7.453 (27) and assigned to PANGO lineages (28) using the Pangolin algorithm (29).

**Identification of phylogenetic clusters.** An initial data set of 4,260 sequences was employed in a maximum likelihood (ML) phylogenetic reconstruction with IQ-TREE v.2.1.3 (30), using a GTR + I + F + Γ4 nucleotide substitution model as selected by ModelFinder (31) to identify possible phylogenetic clusters of VOC Delta in Brazilian territory. The reliability of the obtained tree topology was estimated with the approximate likelihood-ratio test (32). Two rounds of ML analysis were performed to improve the presentation of the results and the resolution of the inferred tree. After the first one, all statistically supported phylogenetic clusters (aLRT, ≥75) without Brazilian sequences were removed. Brazilian phylogenetic clusters in the ML tree were defined as any statistically supported (aLRT ≥ 75) group of more than one sequence mostly (≥75%) originated in one of the country's five regions.

**Phylogeographic analysis.** Two major Delta transmission clusters detected in the southern and southeastern regions were selected for further analysis alongside a few of their basal non-Brazilian sequences. To trace back their most recent common ancestor (MRCA) and reconstruct their spatial diffusion pattern, a time-scaled phylogenetic tree was inferred in the Bayesian Markov chain Monte Carlo (MCMC) approach as implemented in the software package BEAST v.1.10.4 (33, 34) with BEAGLE (35) to improve run-time efficiency. Bayesian MCMC analyses were performed using a strict molecular clock model, a constant prior distribution (probability distribution that express prior belief about this quantity before evidence is considered) on the substitution rate ($8 \times 10^{-4}$ to $10 \times 10^{-4}$ substitutions per site per year), and the nonparametric Bayesian skyline model as a coalescent tree prior (36). Migration events were reconstructed using a reversible discrete phylogeographic model (37) with a continuous time Markov chain (CTMC) rate reference prior (38). MCMC chains were run for $200 \times 10^6$ generations, and convergence and uncertainty of parameter estimates were assessed by calculating the effective sample size (ESS) and 95% highest probability density (HPD) values with Tracer v.1.7.1 (39). Convergence of parameters was considered with an ESS of ≥200. The maximum clade credibility (MCC) trees were summarized with TreeAnnotator v.1.10.4 (34) and visualized with FigTree v.1.4.4 (40).

**Molecular signatures.** The Nextclade algorithm (41) was used to access the molecular composition of the Brazilian genomes. All nonsynonymous substitutions and deletions detected in the majority of sequences (≥75%) were represented in a heatmap generated with the ggplot2 package (42).

## SUPPLEMENTAL MATERIAL

Supplemental material is available online only.

**SUPPLEMENTAL FILE 1**, PDF file, 0.5 MB.

## ACKNOWLEDGMENTS

We thank all the health care workers and scientists who have worked hard to deal with this pandemic threat, the GISAID team, and all the EpiCoV database submitters. GISAID acknowledgment table containing sequences used in this study is shown in the supplemental material. We also appreciate the support of the FIOCRUZ COVID-19 Genomic Surveillance Network (http://www.genomahcov.fiocruz.br/) members, the Respiratory Viruses Genomic Surveillance Network of the General Laboratory Coordination (CGLab), the Brazilian Ministry of Health (MoH), the Brazilian Central Laboratory States (LACENs), and the Amazonas surveillance teams for the partnership in the viral surveillance in Brazil.

Financial support was provided by Fundação de Amparo à Pesquisa do Estado do Amazonas (FAPEAM) (PCTI-EmergeSaúde/AM call 005/2020 and Rede Genômica de Vigilância em Saúde-REGESAM), Conselho Nacional de Desenvolvimento Científico e Tecnológico (CNPq) (grant 402457/2020-0); CNPq/Ministério da Ciência, Tecnologia, Inovações e Comunicações/Ministério da Saúde (MS/FNDCT/SCTIE/Decit) (grant 403276/2020-9), Departamento da Ciência e Tecnologia (DECIT), Ministério da Saúde, Inova Fiocruz/Fundação Oswaldo Cruz (grants VPPCB-007-FIO-18-2-30 and VPPCB-005-FIO-20-2-87), INCT-FCx (465259/2014-6), and Fundação Carlos Chagas Filho de Amparo à Pesquisa do Estado do Rio de Janeiro (FAPERJ) (26/210.196/2020). This work was also supported by the Pan American Health Organization (PAHO), Brazil Country Office. F.G.N., G.L.W., G.B., and M.M.S. are supported by the CNPq through their productivity research fellowships (306146/2017-7, 303902/2019-1, 302317/2017-1, and 313403/2018-0, respectively). G.B. is also funded by FAPERJ (grant E-26/202.896/2018).

I.A., F.G.N., T.G., E.D., M.M.S., G.B., and P.C.R. conceived and designed the study. F.G.N., F.M., H.F., and G.L.W. collected the samples and worked on sequence protocols. L.R.A., E.C.P., T.M.M.V., A.S.R., and R.S.L. worked on sequencing protocols. I.A. and P.C.R. performed the phylogenetic and phylodynamics inferences. I.A., G.B., and P.C.R. wrote the first draft of the manuscript. All authors reviewed and approved the final version of the manuscript.

The participants in the Fiocruz Genomic Surveillance Network in Brazil are Carlos Leonardo Araújo, Cleber Furtado Akesenen, Fernando Braga Stehling Dias, Igor Oliveira Duarte, Jamille Mendes Bezerra, Joaquim Cesar Sousa, Jr., Pedro Miguel Carneiro Jerônimo, Suzana Almeida Porto, Thaís de Oliveira Costa, Thais Ferreira de Oliveira, Ticiane Cavalcante de Souza, Veridiana Pessoa Miyajima (Analytical Competence Molecular Epidemiology Laboratory [ACME], Fundação Oswaldo Cruz [FIOCRUZ], Ceará); Acacia Lourenço Francisco Nasr, Ana Carolina De la Vechia, Rosana Aparecida Piler, Tatiane Motta Huggler (Divisão de Vigilância de Doenças Transmissíveis, Secretaria Estadual de Saúde do Paraná, Paraná); Cristiano Fernandes (Fundação de Vigilância em Saúde do Amazonas, Dra. Rosemary Costa Pinto, Amazonas); Marcelo Gomes (Grupo de Métodos Analíticos em Vigilância Epidemiológica, Programa de Computação Científica [PROCC], FIOCRUZ, Rio de Janeiro); Adriano Abbud, Katia Oliveira Correa (Instituto Adolfo Lutz, São Paulo); Alexandre Freitas da Silva, Antonio Marinho da Silva Neto, Cássia Docena, Filipe Zimmer Dezordi, Gustavo Barbosa de Lima, Laís Ceschini Machado, Lilian Caroliny Amorim Silva, Marcelo Henrique Santos Paiva, Matheus Filgueira Bezerra, Raul Emídio de Lima (Instituto Aggeu Magalhães, FIOCRUZ, Pernambuco); Andreia Akemi Suzukawa, Mauro de Medeiros Oliveira, Michelle Orane Schemberger (Instituto Carlos Chagas [ICC], FIOCRUZ, Paraná, Brasil); Beatriz Grinsztejn, Patricia Brasil, Valdiléa G. Veloso (Instituto Nacional de Infectologia Evandro Chagas [INI], FIOCRUZ, Rio de Janeiro); Felicidade Pereira (Laboratorio Central de Saúde Pública da Bahia [LACEN-BA], Bahia); Dalane Loudal Florentino Teixeira, Haline Barroso (LACEN da Paraíba [LACEN-PB], Paraíba); Anderson Brandao Leite (LACEN de Alagoas [LACEN-AL], Alagoas); Vinicius Lemes da Silva (LACEN de Goiás [LACEN-GO], Goiás); André Felipe Leal Bernardes, Felipe Campos de Melo Iani (LACEN de Minas Gerais [LACEN-MG], Minas Gerais); Irina

Riediger, Maria do Carmo Debur (LACEN de Paraná [LACEN-PR], Paraná); Themis Rocha (LACEN de Sergipe [LACEN-RN], Rio Grande do Norte); Andreia Santos Costa, Lindomar dos Anjos Silva (LACEN do Amapá [LACEN-AP], Amapá); Tirza Peixoto Mattos (LACEN do Amazonas [LACEN-AM], Amazonas; Ana Barjud Marques, Maximo Liana Perdigão Mello, Vania Angelica Feitosa Viana (LACEN do Ceará [LACEN-CE], Ceará); Rodrigo Ribeiro Rodrigues (LACEN do Espirito Santo [LACEN-ES], Espírito Santo); Darcita Buerger Rovaris, Sandra Bianchini Fernandes (LACEN do Estado de Santa Catarina [LACEN-SC], Santa Catarina); Lidio Gonçalves Lima Neto (LACEN do Maranhão [LACEN-MA], Maranhão); Valnete Andrade (LACEN do Pará [LACEN-PA], Pará); Andrea Cony Cavalcanti (LACEN do Rio de Janeiro [LACEN-RJ], Rio de Janeiro); Richard Steiner Salvato Tatiana Schäffer Gregianini (LACEN do Rio Grande do Sul [LACEN-RS] Rio Grande do Sul); Jucimária Dantas Galvão (LACEN do Tocantins [LACEN-TO], Tocantins); Ágatha Costa, André de Lima Guerra Corado, Fernanda Nascimento, George Silva, Karina Pessoa, Luciana Fé Gonçalves, Maria Júlia Brandão, Matilde Mejía, Michele Silva de Jesus, Valdinete Alves Nascimento, Victor Souza (Laboratório de Ecologia de Doenças Transmissíveis na Amazônia [EDTA], Instituto Leônidas e Maria Deane, FIOCRUZ, Amazonas); Bruna Mendonça da Silva, Fernando do Couto Motta, Jéssica de Macedo Carvalho, Larissa Macedo Pinto (Laboratório de Vírus Respiratórios e Sarampo, Instituto Oswaldo Cruz [IOC], FIOCRUZ, Rio de Janeiro); Fernando Vinhal (Laboratório HLAGYN, Goiás); Isabela de Lucena Heráclio Morgana de Freitas Caraciolo, Roberta Mendes Abreu Silva, Silvio Rodrigues de Almeida, Thayna Karoline Sousa Silva (Programa de Treinamento em Epidemiologia Aplicada [EpiSUS-Avançado], Ministério da Saúde); Alessandro Álvares Magalhães, Érika Lopes Rocha Batista (Secretaria de Saúde de Aparecida de Goiânia, Goiás); Greice Madeleine Ikeda do Carmo, Janaína Sallas, Walquiria Aparecida Almeida (Secretaria de Vigilância em Saúde, Ministério da Saúde); Marcio Garcia, Cecilia Leite Costa, Eduardo Ruback dos Santos (Secretaria de Vigilância em Saúde, Secretaria Municipal de Saúde-Rio de Janeiro, Brasil Unidade de Apoio Diagnostico [UNADIG], FIOCRUZ, Ceará); João Felipe Bezerra (Universidade Federal da Paraíba [UFPB], Paraíba).

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
