## [Reviewer comments · Microbiology Spectrum]

Microbiology Spectrum

Emergence and Spread of the SARS-CoV-2 Variant of Concern Delta Across Different Brazilian Regions

Ighor Arantes, Felipe Gomes Naveca, Tiago Gräf, COVID-19 Fiocruz Genomic Surveillance Network, Fabio Miyajima, Helisson Faoro, Gabriel Wallau, Edson Delatorre, Luciana Appolinario, Elisa Pereira, Tainá Venas, Alice Rocha, Renata Lopes, Marilda Mendonça Siqueira, Gonzalo Bello, and Paola Resende

Corresponding Author(s): Paola Resende, Laboratory of Respiratory Viruses and Measles, Oswaldo Cruz Institute (IOC), FIOCRUZ, Rio de Janeiro, Brazil

Review Timeline:

Submission Date:	January 20, 2022
Editorial Decision:	February 18, 2022
Revision Received:	July 7, 2022
Accepted:	July 12, 2022

Editor: Juan Ludert

Reviewer(s): Disclosure of reviewer identity is with reference to reviewer comments included in decision letter(s). The following individuals involved in review of your submission have agreed to reveal their identity: Selene Zarate (Reviewer #1)

Transaction Report:

DOI: <https://doi.org/10.1128/spectrum.02641-21>

February 18, 2022

Dr. Paola Cristina Resende
Laboratory of Respiratory Viruses and Measles, Oswaldo Cruz Institute (IOC), FIOCRUZ, Rio de Janeiro, Brazil
AV BRASIL, 4365
MANGUINHOS
RIO DE JANEIRO, RJ 21040900
Brazil

Re: Spectrum02641-21 (Emergence and Spread of the SARS-CoV-2 Variant of Concern Delta Across Different Brazilian Regions)

Dear Dr. Paola Cristina Resende:

Thank you for submitting your manuscript to Microbiology Spectrum. Your manuscript was revised by two experts in the field and they both agree that the manuscript may be suitable for publication after revision. Please note that both reviewers mention the necessity to deposit in a public database, the generated sequences. When submitting the revised version of your paper, please provide (1) point-by-point responses to the issues raised by the reviewers as file type "Response to Reviewers," not in your cover letter, and (2) a PDF file that indicates the changes from the original submission (by highlighting or underlining the changes) as file type "Marked Up Manuscript - For Review Only". Please use this link to submit your revised manuscript - we strongly recommend that you submit your paper within the next 60 days or reach out to me. Detailed instructions on submitting your revised paper are below.

Link Not Available

Sincerely,

Juan E. Ludert

Journals Department
Reviewer comments:

Reviewer #1 (Public repository details (Required)):

The study includes the sequencing of viral samples. Therefore, those sequences must be deposited in a public database

Reviewer #1 (Comments for the Author):

The work aims to characterize the spread and circulation of the Delta variant in Brazil. The study is comprehensive, highlights regional differences, and determines specific mutations in Brazil's lineages. There are, however, some concerns about some of the strategies utilized, particularly in the subsampling.

Major concerns:

The Delta dataset's downsizing strategy consists of automatically eliminating duplicates and keeping the sequences more similar to those from Brazil. However, there is a concern about the evenness of this subset's temporal and geographical distribution, mainly since it is used to infer the geographical origin of the main sublineages. It will be helpful to show the available Delta sequences in the database by date and region and consider enriching the representation of those states with fewer samples.

Given the unevenness of the sampling, it will also be helpful to present the total number of sequences available from Brazil in this period to assess if the contribution of other circulating variants was more significant in the north of Brazil or if a lack of sampling is to blame.

Given the sampling biases, what are the effective sample sizes reported by Tracer? Are they within the accepted values?

Minor concerns:

There are some issues with the language in the abstract, where the verbs used are not always exact; for instance, the use of the verb "born" to mean originated should be changed

Figure 1 legend: please correct "their dimension have the same scholar scheme"

Reviewer #2 (Public repository details (Required)):

sars-cov-2 genomes

Reviewer #2 (Comments for the Author):

The authors explored the emergence and spread of Delta in different Brazilian regions, and identified three major clusters. Although this work is important and the methods are solid, there are a few points that should be addressed in order to make the manuscript stronger.

As the authors mentioned in the introduction, until the summer of 2021 gamma was the main variant spreading through the population in Brazil. It would be of great help for the readers of this article to see a plot of variants in Brazil overtime, and show when exactly gamma was displaced by delta. This would give an overview of the state of the population when delta came to play.

Moreover, linked to gamma infections, it is conceivable that the population was infected at a high rate with gamma when delta came through. What is the seropositivity in the population at the time of delta surge? Vaccination status and number cases? This information need to be there.

It would also be very useful to add to the figures how many genomes have been sequenced from each region represented and to be specific whether there is a geographic bias in sampling and how the authors overcome that limitation.

The authors should also explore the possibility of these genomes to be linked to external introductions in Brazil, for that I would suggest the authors to blast their genomes against GISAID non-Brazil genomes and collect the ones that are the most similar before and after the collection date of each genome within a month. That way it would be possible to see how these clusters are local or due to importations. This is an important analysis that needs to be made in order to be sure that the clusters the authors found are correct.

Are the clusters specific genomic signatures indication of new delta sub-lineages?

Staff Comments:

Preparing Revision Guidelines

Please return the manuscript within 60 days; if you cannot complete the modification within this time period, please contact me. If you do not wish to modify the manuscript and prefer to submit it to another journal, please notify me of your decision immediately so that the manuscript may be formally withdrawn from consideration by Microbiology Spectrum.

Reviewer comments

Reviewer #1 (Public repository details (Required)):

The study includes the sequencing of viral samples. Therefore, those sequences must be deposited in a public database

All Brazilian sequences newly generated by the COVID-19 Fiocruz Genomic Surveillance network, were submitted to the GISAID database (<https://www.gisaid.org/>) in advance to the original submission, and their accession codes were listed in the APPENDIX 2. The other sequences, inside and outside Brazil were also downloaded from GISAID.

Reviewer #1 (Comments for the Author):

1. The work aims to characterize the spread and circulation of the Delta variant in Brazil. The study is comprehensive, highlights regional differences, and determines specific mutations in Brazil's lineages. There are, however, some concerns about some of the strategies utilized, particularly in the subsampling.

Major concerns:

2. The Delta dataset's downsizing strategy consists of automatically eliminating duplicates and keeping the sequences more similar to those from Brazil. However, there is a concern about the evenness of this subset's temporal and geographical distribution, mainly since it is used to infer the geographical origin of the main sublineages. It will be helpful to show the available Delta sequences in the database by date and region and consider enriching the representation of those states with fewer samples.

The downsizing of our dataset by the elimination of duplicated genomes was applied only to non-Brazilian sequences while those from Brazil available until 2021-09-07 were all kept, and removed only if falling in one or more of the two cases of exclusion, I) absence of collection date, and II) $N_s > 3\%$. We agree with the reviewer's observation that the downsampling of non-Brazilian sequences might bias the inferred epidemiological links between the three identified main clades (BR-I, BR-II and BR-III) and sequences sampled outside Brazil. Therefore, the presentation of the results was limited to the reconstructed ancestral location of the clades' MRCA already in the country, where the dataset was kept in its entirety. The text was changed in the newly submitted version to clarify the actual methodology.

3. Given the unevenness of the sampling, it will also be helpful to present the total number of sequences available from Brazil in this period to assess if the contribution of other circulating variants was more significant in the north of Brazil or if a lack of sampling is to blame.

In order to clarify the relative prevalence of the differently circulating variants across time, we added a new Fig 1, showing the contribution of Delta (B.1.617.2 + AY), Gamma (P.1 + P.1*) and other circulating variants in the country's available genome sequences. In Fig 1 A-F is shown the absolute number of SARS-CoV-2 positive cases by region and the relative prevalence of VOCs Gamma and Delta and of other variants. The data used in this representation is available at our institution website (in english): <http://www.genomahcov.fiocruz.br/dashboard-en/>. In the graphs, it is perceivable that Brazil's northern region's small contribution to the Delta epidemic during the period considered in the present study is not due to a particularly low sampling in the period as it is coincidental with a marked reduction in the total number of positive cases in those states.*

4. Given the sampling biases, what are the effective sample sizes reported by Tracer? Are they within the accepted values?

All parameters of the BSKL model analysis in Beast 1.10 indeed showed ESS values above 200. To acquiesce the reviewer's concern, the methodology section was updated to clarify this aspect.

Minor concerns:

5. There are some issues with the language in the abstract, where the verbs used are not always exact; for instance, the use of the verb "born" to mean originated should be changed

The abstract text was revised in compliance with the reviewer's suggestion.

6. Figure 1 legend: please correct "their dimension have the same scholar scheme".

The legend was revised in compliance with the reviewer's suggestion.

Reviewer #2 (Public repository details (Required)):

1. sars-cov-2 genomes

All Brazilian sequences newly generated by the COVID-19 Fiocruz Genomic Surveillance network, were submitted to the GISAID database (<https://www.gisaid.org/>) in advance to the original submission, and their accession codes were listed in the APPENDIX 2. The other sequences, inside and outside Brazil were also downloaded from GISAID.

Reviewer #2 (Comments for the Author):

The authors explored the emergence and spread of Delta in different Brazilian regions, and identified three major clusters. Although this work is important and the methods are solid, there are a few points that should be addressed in order to make the manuscript stronger.

1. As the authors mentioned in the introduction, until the summer of 2021 gamma was the main variant spreading through the population in Brazil. It would be of great help for the readers of this article to see a plot of variants in Brazil overtime, and show when exactly gamma was displaced by delta. This would give an overview of the state of the population when delta came to play.

In order to clarify the relative prevalence of the different circulating variants across time in Brazil, we added Figure 1 showing the contribution of Delta (B.1.617.2 + AY), Gamma (P.1 + P.1*) and other circulating variants to the country's available genome sequences. The data used in this representation is available at our institution website (in english): <http://www.genomahcov.fiocruz.br/dashboard-en/>.*

2. Moreover, linked to gamma infections, it is conceivable that the population was infected at a high rate with gamma when delta came through. What is the seropositivity in the population at the time of delta surge? Vaccination status and number cases? This information needs to be there.

Even though a comprehensive analysis of the Brazilian population's seropositivity by the time of introduction of Delta is unavailable, we added a Supplementary Figure 1 that describes the vaccination status of the population, showing the evolution of individuals with at least one dose as well as those fully vaccinated. The Discussion section was also modified to include this information.

3. It would also be very useful to add to the figures how many genomes have been sequenced from each region represented and to be specific whether there is a geographic bias in sampling and how the authors overcome that limitation.

The sequences listed in Table 1 represent all Delta genomes available from Brazil until 2021-09-07, and clearly the South (10.38%) and Southeast (83.17%) regions are the most represented in the overall dataset. Nonetheless, this observation is not the product of a bias in the sequences used in the study. As can be seen in the graphs represented in Figure 1, there is a homogenous geographical disparity in contribution to the country's SARS-CoV-2 genomes across time and a roughly comparable evolution in the contribution of VOC's Gamma and Delta across time in the five geographic regions'.

4. The authors should also explore the possibility of these genomes to be linked to external introductions in Brazil, for that I would suggest the authors to blast their genomes against GISAID non-Brazil genomes and collect the ones that are the most similar before and after the collection date of each genome within a month. That way it would be possible to see how these clusters are local or due to importations. This is an important analysis that needs to be made in order to be sure that the clusters the authors found are correct.

The way in which the dataset composition was described in the original submission was ambiguous and didn't reflect the actual process, having been updated in the newly submitted version. Therefore, it's possible the reviewer concern has already been addressed. All Delta variant sequences available until 2021-09-07 (n = 948,925) at GISAID (<https://www.gisaid.org/>) were downloaded. All Brazilian sequences were kept in the study, being removed only if falling in one or more of the two cases of exclusion, I) absence of collection date and II) Ns > 3%. Sequences outside Brazil, however, were downsampled in three stages. In the first one, identical sequences were removed using the CD-hit software. In the second one, all Brazilian sequences were employed as a query in a local blast search, as suggested by the reviewer, and we selected the 10 non-Brazilian sequences most similar to the Brazilian queries. The sequences kept then were used to infer an initial ML tree (not shown) and in the third and final stage of the downsampling, clades not including a Brazilian sample were removed, leading to the dataset used in the final ML tree (Figure 2). Therefore, all Brazilian sequences have been exposed by a blast search to the Delta lineage worldwide diversity, increasing the chances, as seen in the statistical support of the identified clades, that their association in the ML topology was not a sampling artifact.

5. Are the clusters specific genomic signatures indication of new delta sub-lineages?

The identified signatures in the three clades in fact characterize them as Delta sub-lineages. BR-I was composed of lineages AY.99/AY.99.1/AY.99.2, BR-II of lineage AY.101, and BR-III of lineage AY.46.3. Given the that lineages AY.101 and AY.46.3 have the absolute majority of their sequences (> 95%) in Brazil, and that no other sequences in both lineages were places at the root of their ML trees, the paper was changed in several

places in order to change the used nomenclature, as BR-II and BR-III were abandoned in favor of the lineages themselves.

July 12, 2022

Dr. Paola Cristina Resende
Laboratory of Respiratory Viruses and Measles, Oswaldo Cruz Institute (IOC), FIOCRUZ, Rio de Janeiro, Brazil
AV BRASIL, 4365
MANGUINHOS
RIO DE JANEIRO, RJ 21040900
Brazil

Re: Spectrum02641-21R1 (Emergence and Spread of the SARS-CoV-2 Variant of Concern Delta Across Different Brazilian Regions)

Dear Dr. Paola Cristina Resende:

I am glad to inform you that your manuscript has been accepted, and I am forwarding it to the ASM Journals Department for publication. You will be notified when your proofs are ready to be viewed.

Sincerely,

Juan E. Ludert
Editor, Microbiology Spectrum
